# Expanding Access to Optically Active Non-Steroidal Anti-Inflammatory Drugs via Lipase-Catalyzed KR of Racemic Acids Using Trialkyl Orthoesters as Irreversible Alkoxy Group Donors

Beata Zdun [1], Piotr Cieśla [1], Jan Kutner [2] and Paweł Borowiecki [1,*]

[1] Laboratory of Biocatalysis and Biotransformation, Department of Drugs Technology and Biotechnology, Warsaw University of Technology, Koszykowa St. 75, 00-662 Warsaw, Poland; beata.zdun.dokt@pw.edu.pl (B.Z.); pciesla16@wp.pl (P.C.)

[2] Laboratory for Structural and Biochemical Research, Biological and Chemical Research Centre, Department of Chemistry, University of Warsaw, Żwirki and Wigury St. 101, 02-089 Warsaw, Poland; j.kutner@uw.edu.pl

* Correspondence: pawel.borowiecki@pw.edu.pl; Tel.: +48-(22)-234-7677

**Abstract:** Studies into the enzymatic kinetic resolution (EKR) of 2-arylpropanoic acids ('profens'), as the active pharmaceutical ingredients (APIs) of blockbuster non-steroidal anti-inflammatory drugs (NSAIDs), by using various trialkyl orthoesters as irreversible alkoxy group donors in organic media, were performed. The enzymatic reactions of target substrates were optimized using several different immobilized preparations of lipase type B from the yeast *Candida antarctica* (CAL-B). The influence of crucial parameters, including the type of enzyme and alkoxy agent, as well as the nature of the organic co-solvent and time of the process on the conversion and enantioselectivity of the enzymatic kinetic resolution, is described. The optimal EKR procedure for the racemic profens consisted of a Novozym 435-STREM lipase preparation suspended in a mixture of 3 equiv of trimethyl or triethyl orthoacetate as alkoxy donor and toluene or *n*-hexane as co-solvent, depending on the employed racemic NSAIDs. The reported biocatalytic system provided optically active products with moderate-to-good enantioselectivity upon esterification lasting for 7–48 h, with most promising results in terms of enantiomeric purity of the pharmacologically active enantiomers of title APIs obtained on the analytical scale for: (*S*)-flurbiprofen (97% ee), (*S*)-ibuprofen (91% ee), (*S*)-ketoprofen (69% ee), and (*S*)-naproxen (63% ee), respectively. In turn, the employment of optimal conditions on a preparative-scale enabled us to obtain the (*S*)-enantiomers of: flurbiprofen in 28% yield and 97% ee, ibuprofen in 45% yield and 56% ee, (*S*)-ketoprofen in 23% yield and 69% ee, and naproxen in 42% yield and 57% ee, respectively. The devised method turned out to be inefficient toward racemic etodolac regardless of the lipase and alkoxy group donor used, proving that it is unsuitable for carboxylic acids possessing tertiary chiral centers.

**Keywords:** biocatalysis; lipases; kinetic resolution; chiral 2-arylpropanoic acids; esterification; non-steroidal anti-inflammatory drugs; trialkyl orthoesters

## 1. Introduction

The 2-arylpropionic acid derivatives, commonly termed 'profens', are an essential group of non-steroidal anti-inflammatory drugs (NSAIDs) that are used for the symptomatic treatment of various forms of arthritis (i.e., rheumatoid arthritis, osteoarthritis, and ankylosing spondylitis). In most cases, the (*S*)-enantiomers of profens are pharmacologically active due to their principal effectiveness in inhibiting cyclooxygenases (COXs)—the enzymes responsible for the synthesis of prostaglandins and other mediators of inflammatory responses obtainable from arachidonic acid. The inhibitory properties of (*S*)-NSAIDs lie at the heart of their anti-inflammatory, analgesic, and antipyretic action in vivo [1,2].

For example, (*S*)-naproxen is ca. 28-fold more potent than the (*R*)-enantiomer [3], which is also highly hepatotoxic and increases the burden on renal clearance. Due to these facts, naproxen is the only NSAID that must be administrated as a single stereochemically pure (*S*)-enantiomer. In addition, the (*S*)-ibuprofen (also known as 'dexibuprofen') is over 100-fold more active as an inhibitor of cyclooxygenase 1 (COX-1) isoenzyme than its (*R*)-counterpart [4–6]. Interestingly, (*R*)-ibuprofen exhibits 'metabolic chiral inversion' into pharmacologically active antipodes; nevertheless, the efficiency of this transformation in humans is in the range of 35–70% and depends on the condition of the liver and the intake of other drugs [7–10]. In turn, in the case of (*R*)-flurbiprofen and (*R*)-ketoprofen, chiral metabolic inversion of the distomers into their opposed (*S*)-enantiomeric forms is limited to a maximum of 10% [11,12]. It is worth noting that only (*S*)-flurbiprofen exhibits inhibitory potency towards prostaglandin biosynthesis [13] and antinociceptive activity [14]. Similarly, (*S*)-ketoprofen is 2–4 times more potent than the racemate and provides reduced gastric irritation and improved tolerability [15]. Low metabolic bioconversion of the distomers into the eutomers in vivo, as well as high stereospecificity of action of the afore-mentioned NSAIDs, caused that for a therapeutic benefit, the (*R*)-ibuprofen and (*R*)-ketoprofen were submitted to 'chiral switch' [16,17]. Such clinical approaches resulted in the introduction of dexibuprofen (manufactured under the tradenames Seractil® or DexOprifen®) [18] and dexketoprofen (available under the tradenames Keral®, Enantyum® or Dolmen®) [19] into therapy for rheumatic diseases in 1994 and 1998, respectively.

Over the last few decades, biocatalytic methods have emerged as an indispensable and versatile tool for the asymmetric synthesis of optically active NSAIDs (Figure 1). Among them, the noteworthy examples are (**1**) the enantioselective hydrolytic kinetic resolution (KR) of the appropriate racemic esters using lipases from *Candida rugosa* (CRL, formerly *Candida cylindracea* (CLL)) [20–26], pig pancreas (PPL) [27], or engineered *Yarrowia lipolytica* (Lip2p) [28], respectively. Moreover, carboxylesterase NP [29], *Pseudomonas fluorescens* MTCCB0015 cell-free extract [30], and whole cells of *Trichosporon* sp. [31], were also used as potent biocatalysts for hydrolytic KR of racemic NSAID-esters. Notably, when (**2**) hydrolytic KR was attempted toward the corresponding racemic NSAID-azolides, lipase B from *Candida antarctica* (CAL-B) was superior to other biocatalysts and catalyzed the reaction with the opposite (*R*)-stereopreference when compared to the afore-mentioned enzymes used in hydrolytic KR [32,33]. In turn, the reversed enzymatic KR process, that is (**3**) enantioselective esterification of the appropriate racemic profens using short-chain aliphatic alcohols, is catalyzed mainly by CAL-B [34–38] and rarely by CRL [39]. Another variant of enzymatic kinetic resolution of racemic profens is the (**4**) enantioselective thioesterification of the appropriate 2-arylpropanoic acids catalyzed by (*R*)-selective Japanese firefly luciferase from *Luciola lateralis* (LUC-H) using coenzyme A (CoASH) as a thioacetyl donor [40].

In search of deracemization and/or desymmetrization processes that are more efficient than classical kinetic resolutions, which are theoretically limited to 50% yield, (**5**) hydrolytic dynamic kinetic resolution (DKR) of methyl ibuprofen ester was developed using *Candida rugosa* lipase suspended in a mixture of an aqueous buffer (pH 9.8)/DMSO (4:1, *v*/*v*) [41]. Ohta et al. [42] developed (**6**) asymmetric decarboxylation of 2-(2-fluoro-4-biphenylyl)-2-methylmalonic acid catalyzed by (*R*)-selective arylmalonate decarboxylase (AMDase). Unfortunately, the AMDase-catalyzed decarboxylation of the aforementioned malonic acid derivative leads to undesired (*R*)-flurbiprofen, which significantly limits the synthetic value of this method. This limitation has been overcome in recent years by Enoki et al. [43], who reported the application of the first engineered (*S*)-selective AMDase useful in the synthesis of (*S*)-profens. Later on, (**7**) a dynamic reductive kinetic resolution (DYRKR) of the corresponding 2-arylpropanals into (2*S*)-2-arylpropanols (valuable intermediates in the synthesis of (*S*)-profens) with horse liver alcohol dehydrogenase (HLADH) [44,45] or recombinant ADH from an archaeal hyperthermophile *Sulfolobus solfataricus* (SsADH-10) [46] were reported. With the expansion of (**8**), the HLADH catalytic potency toward prochiral profenals coupled with in situ oxidation of the formed (*S*)-profenols by laccase

from *Trametes versicolor* (L*Tv*) and the stable free radical 2,2,6,6-tetramethylpiperidine-*N*-oxyl (TEMPO), leading to (*S*)-profens, was demonstrated by Giacomini et al. [47].

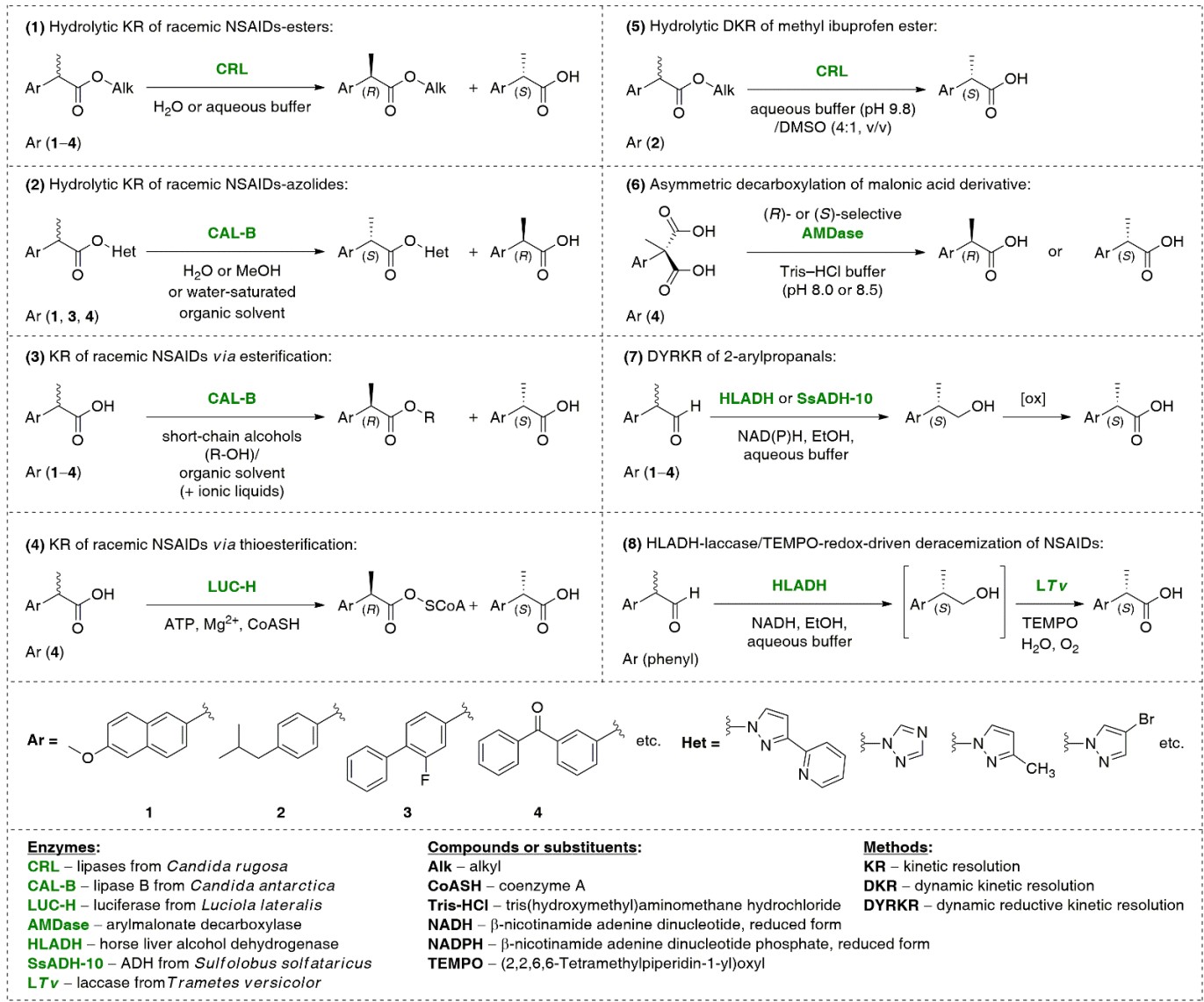

**Figure 1.** Overview of biocatalytic strategies for the synthesis of enantiomeric non-steroidal anti-inflammatory drugs (NSAIDs).

Although multiple routes toward enantiomeric NSAIDs have already been devised, there is still an urgent need to develop cost-efficient and sustainable methodologies for synthetic contingency plans to produce these APIs on a large scale in case of market shortages. Among the biocatalytic attempts, lipase-catalyzed hydrolytic KR of racemic alkyl 2-arylpropanoates in aqueous buffers and/or enantioselective esterification of the respective carboxylic acids employing low-molecular-weight alcohols (i.e., methanol and ethanol) as alkoxy donors are the most common. However, hydrolytic KR suffers from tedious extractive workups, while transesterification using short-chain alcohols exhibits detrimental effects on the stability of lipases (i.e., deactivation of the enzymes leading to low reaction rates). Therefore, more efficient biocatalytic methods to prepare enantiomerically enriched (*S*)-NSAIDs are required. With the aim to expand the synthetic toolbox for the preparation of title APIs, we present the extension of classical lipase-catalyzed enantioselective esterification of the racemic profens using kinetic resolution conditions and trialkyl orthoesters as irreversible donors of the alkoxy group in organic solvents. The systematic

optimization studies reported in this paper show great potential for applying orthoesters in lipase-catalyzed kinetic resolutions of racemic compounds possessing carboxylic acid moiety.

## 2. Results and Discussion

Herein, we report on the development of a chemoenzymatic route toward enantiomerically enriched dextro-isomers of non-steroidal anti-inflammatory drugs (NSAIDs), namely (*S*)-(+)-2-(6-methoxynaphthalen-2-yl)propanoic acid [naproxen, (*S*)-(+)-**1**], (*S*)-(+)-2-(4-(2-methylpropyl)phenyl)propanoic acid [ibuprofen, (*S*)-(+)-**2**], (*S*)-(+)-2-(3-benzoylphenyl) propanoic acid [ketoprofen, (*S*)-(+)-**3**] and (*S*)-(+)-2-(2-fluorobiphenyl-4-yl)propanoic acid [flurbiprofen, (*S*)-(+)-**4**], respectively. The synthetic pathway for the asymmetric synthesis of non-racemic title APIs consists of lipase-catalyzed enantioselective esterification of the appropriate racemic 2-arylpropanoic acids carried out under kinetically controlled conditions in organic solvents (Scheme 1).

**Scheme 1.** Lipase-catalyzed enantioselective resolution of racemic NSAIDs *rac*-**1**–**5** in organic solvents using trialkyl orthoesters **Alk-D1**–**D4** as irreversible alkoxy group donors.

Most enzymes, including lipases, are inactivated by low-molecular-weight alcohols, particularly methanol and ethanol [48–51]. The evidence of a detrimental effect of the short-chain alcohols on lipases was also investigated during the esterification of ibuprofen [52]. Moreover, the latest studies on the most common heterogeneous biocatalyst used in the enantioselective esterification of racemic alcohols and acids, that is lipase B from *Candida antarctica* adsorbed on polymethacrylate beads (Novozym 435®), confirmed that the short-chain alcohols induce not only conformational changes leading to CAL-B aggregation, but also modifies the texture of the solid support, promoting the enzyme release [53].

Therefore, to overcome the drawbacks associated with short-chain alcohols, alternatively, long-chain fatty alcohols are employed [54,55]. Although inhibition phenomena are limited in this case, the rates of the enzymatic reactions are significantly lower because more

bulky molecules diffuse at the lipase active site less quickly. At the same time, the nucleophilic character of alcohol decreases as the length of its aliphatic chain increases. Therefore, another proven strategy to eliminate the adverse effects of the excessive short-chain alcohols is their stepwise addition [56] or solubilization in *tert*-butanol [57]. Regardless of the modifications used, the esterifications involving alcohols are disadvantageous because they do not guarantee the irreversibility of the process due to water formation and an unfavorable reaction's equilibrium.

In turn, orthoesters, in principle, offer a significant advantage over the traditional alkoxy donors in that lipase can catalyze esterification without enzyme inactivation, and the irreversibility of the reaction is preserved due to the consumption of the in situ formed water, which at the same time release the nucleophilic alcohol for the esterification. The first example of the application of orthoesters in direct irreversible lipase-catalyzed enantioselective esterification of racemic profens was reported by Nicolosi et al. [58,59] However, this approach has been limited only to the resolution of ketoprofen enantiomers and a single orthoformate as an alkoxy group donor. In the last few years, the utility of this method has also been presented toward other α-substituted chiral acids (i.e., 3-phenyl-4-pentenoic acid) [60–63] and β-hydroxy acids (i.e., 3-hydroxy-3-(aryl)propanoic acids) [64], respectively. Considering all the pros and cons of the enzymatic esterifications of carboxylic acids, we decided to expand upon a valuable strategy of using trialkyl orthoesters as irreversible donors of alcoholate groups to the lipase-catalyzed esterification of racemic profens other than ketoprofen.

The study on enzymatic kinetic resolution (EKR) of profens was initiated by screening the commercially available enzymes in a model reaction with two benchmark racemic substrates—naproxen (*rac*-**1**) and ibuprofen (*rac*-**2**). The enzymatic reactions were conducted on an 87 μmol scale at 40 °C for 24–72 h in toluene (PhCH$_3$) containing 3 equiv of trimethyl orthoacetate (**Alk-D1**), and 10 mg of the enzyme what constituted ca. 50% *w/w* in respect to the appropriate substrate *rac*-**1** and *rac*-**2** (Table 1).

The obtained EKR products were isolated and purified using preparative column chromatography before being subjected to further HPLC analysis. In order to find the most active and enantioselective enzyme preparation for the esterification of both racemic profens *rac*-**1** and *rac*-**2**, a set of 26 different hydrolases was tested (not shown herein). In most cases, poor results were obtained in terms of the conversion of the substrates. Moreover, no reaction progress was detected in the case of lipase from *Candida antarctica* type A (CAL-A, Chirazyme L-5), *Burkholderia cepacia* (Amano PS-IM, PS-Immobead 150, Amano PS), *Pseudomonas fluorescens* (Amano AK), *Thermomyces lanuginosus* (TL-Immobead 150, Lipozyme TL IM), *Rhizomucor miehei* (Lipozyme RM IM), *Alcaligenes* sp. (Chirazyme L-10), *Mucor javanicus* (Amano M, Amano 10 Lipase M), *Rhizopus oryzae* (Amano Lipase F-AP15), *Candida rugosa* (Lipase AY Amano 30, Chirazyme L-3, Lipase Type VII), *Geotrichum candidum* (Chirazyme L-8), *Penicillium camemberti* (Lipase G50 Amano), *Aspergillus niger* (Amano A) and esterase from the porcine liver (PLE).

In turn, the most promising results were obtained with the commercial enzyme preparations that contained lipase type B from *Candida antarctica* (CAL-B). Interestingly, all lipases were employed in the immobilized forms and were appropriately manufactured using different carriers. According to the indications of the respective suppliers, Novozym 435 was immobilized on the macroporous acrylic resin [poly (methyl methacrylate-co-butyl methacrylate)], Lipozyme 435 on Lewatit VP OC 1600, Chirazyme L-2, C-2 on the carrier-fixed 2 (Carrier 2®), Chirazyme L-2, C-3 on the carrier-fixed 3 (Carrier 3®), both CAL-B Sigma L4777 and Novozym 435-STREM on the macroporous acrylic resin, and finally, CAL-B-Immobead 150 on the Immobead 150®. Thanks to immobilized preparations of lipases, the workup and isolation procedures were greatly simplified. Moreover, the immobilized biocatalysts could also be recycled and used in multi-batches operations, potentially improving the economics of a particular biocatalytic technology.

**Table 1.** Enzyme screening for enantioselective esterification of naproxen (*rac*-**1**) and ibuprofen (*rac*-**2**) with trimethyl orthoacetate (**Alk-D1**) under KR-conditions in $PhCH_3$ at 40 °C.

| Entry | Substrate [a] | CAL-B Preparation | Time (h) | Conv. (%) [b] | ee$_s$ (%) [c] | ee$_p$ (%) [c] | $E$ [d] |
|---|---|---|---|---|---|---|---|
| 1 | | Novozym 435 | 72 | 7 | 6 | 75 | 7 |
| 2 | | Lipozyme 435 | 72 | 13 | 13 | 86 | 15 |
| 3 | | Chirazyme L-2, C-2 | 72 | 13 | 11 | 74 | 7 |
| 4 | *rac*-**1** | Chirazyme L-2, C-3 | 72 | 52 | 57 | 53 | 6 |
| 5 | | CAL-B Sigma L4777 | 72 | 8 | 6 | 71 | 6 |
| 6 | | Novozym 435-STREM | 72 | 46 | 55 | 65 | 8 |
| 7 | | CAL-B-Immobead 150 | 72 | 16 | 14 | 76 | 8 |
| 8 | | Novozym 435 | 24 | 65 | 87 | 46 | 7 |
| 9 | | Lipozyme 435 | 24 | 44 | 52 | 65 | 8 |
| 10 | | Chirazyme L-2, C-2 | 24 | 61 | 62 | 39 | 4 |
| 11 | *rac*-**2** | Chirazyme L-2, C-3 | 24 | 46 | 55 | 66 | 8 |
| 12 | | CAL-B Sigma L4777 | 24 | 52 | 57 | 52 | 5 |
| 13 | | Novozym 435-STREM | 24 | 68 | 91 | 42 | 7 |
| 14 | | CAL-B-Immobead 150 | 24 | 55 | 76 | 63 | 10 |

[a] Conditions: *rac*-**1**–**2** 87 μmol, CAL-B 10 mg, $PhCH_3$ 2 mL, trimethyl orthoacetate (**Alk-D1**, 3 equiv), 40 °C, 800 rpm (magnetic stirrer); [b] Calculated from the enantiomeric excess of the unreacted carboxylic acid (ee$_s$) and the formed ester (ee$_p$) according to the formula conv. = ee$_s$/(ee$_s$ + ee$_p$); [c] Determined by chiral HPLC analysis using a (*S*,*S*)-Whelk-O 1 column; [d] Calculated according to Chen et al. [65], using the equation: $E = \{\ln[(1 - \text{conv.})(1 - \text{ee}_s)]\}/\{\ln[(1 - \text{conv.})(1 + \text{ee}_s)]\}$.

The experimental data show that the most enantioselective EKR reactions were catalyzed by Lipozyme 435 (in the case of *rac*-**1**) and by CAL-B-Immobead 150 (in the case of *rac*-**2**). Although Lipozyme 435 exhibited the highest enantioselectivity ($E$ = 15) in the case of the kinetic resolution of *rac*-**1**, the substrate conversion was substantially lower (13% conv.) compared with other CAL-B preparations, except for Novozym 435, Chirazyme L-2, C-2, and CAL-B Sigma L4777, which also catalyzed the reactions rather sluggishly (7–13% conv. after 72 h). When taking into account the enantiomeric purity of the desired (*S*)-NSAIDs, it was apparent that both Chirazyme L-2, C-3, and Novozym 435-STREM were superior to the other biocatalysts used in the EKR of model racemic naproxen (*rac*-**1**), furnishing enantiomerically enriched (*S*)-(+)-**1** with 55–57% ee and 46–52% conv., respectively. In turn, during the screenings of the CAL-B-catalyzed EKR of *rac*-**2**, it turned out that Novozym 435-STREM was the only lipase preparation that led to afford (*S*)-(+)-**2** with 91% ee and 68% conv. Although, in this case, the enantioselectivity factor ($E$) did not exceed the value of 10, which is known as industrially acceptable [66], it was evident that the desired (*S*)-enantiomer with >90% ee could only be obtained with Novozym 435-STREM.

It is also noteworthy that by using various enzyme preparations of the same CAL-B protein, the outcome of the reaction might differ a lot in terms of the % conv. and % ee values. The most visible differences are for both Novozym- and Chirazyme-type preparations. In the example, the lipase provided by STREM® company (i.e., Novozym 435-STREM) turned out to be significantly more potent toward racemic naproxen (*rac*-**1**) than the preparation purchased from Novozymes®. In this regard, Novozym 435-STREM catalyzed EKR of *rac*-**1** with 46% conv., whereas Novozym 435 converted *rac*-**1** with barely 7% conv. Moreover, when comparing both the Chirazyme preparations purchased from Roche®, the conversion of the EKR reactions reached 13% in the case of Chirazyme L-2 and C-2, and 52% in the case of Chirazyme L-2, C-3. Interestingly, when analyzing the results obtained for EKR of ibuprofen (*rac*-**2**), the differences in the outcome of the reactions among the lipases mentioned above were negligible in terms of enantioselectivity and conversions.

This phenomenon suggests that the mutual interactions in the substrate–protein complex are critical and strongly dependent on the immobilized form of the biocatalyst. It is expected that the method of immobilization and the type of carrier are both responsible for conformational changes and the rigidity of the proteins and may be critical in terms of their biocatalytic performances. In turn, the difference in catalytic efficiency of the studied commercial CAL-B preparations might also be attributed to enzyme loading, which is dependent on the surface morphology of the mesoporous supports (i.e., pore size, pore volume, surface area, etc.). However, these aspects decide rather on the catalytic activity of the enzymes and not on their enantioselectivity.

In the next stage, we focused on investigating the effect of various orthoesters **Alk-D1–D4** as alkoxy group donors on the CAL-B-catalyzed KR of the selected profens (Table 2). The evaluation of this issue was critical for our studies from the viewpoint of enantioselectivity of the examined enzymatic processes and the possible variations in stereopreference observed among lipases depending on the orthoester used. In this regard, Ostaszewski et al. [62] demonstrated that a simple change of the alkoxy group donor could reverse the stereochemical course of the EKR of 3-phenyl-4-pentenoic acid when lipase from goose liver acetone powder (GLAP) was used as a biocatalyst.

**Table 2.** Alkoxy group donor (3 equiv) screening for (Novozym 435-STREM)-catalyzed KR of naproxen (*rac*-**1**) and ibuprofen (*rac*-**2**) in PhCH$_3$ at 40 °C.

| Entry | Substrate [a] | Alkoxy Donor | Time (h) | Conv. (%) [b] | ee$_s$ (%) [c] | ee$_p$ (%) [c] | E [d] |
|---|---|---|---|---|---|---|---|
| 1 | | Trimethyl orthoacetate (**Alk-D1**) | 72 | 46 | 55 | 65 | 8 |
| 2 | | Triethyl orthoacetate (**Alk-D2**) | 72 | 17 | 13 | 62 | 5 |
| 3 | | Trimethyl orthobenzoate (**Alk-D3**) | 72 | 68 | 63 | 29 | 3 |
| 4 | | Triethyl orthobenzoate (**Alk-D4**) | 72 | 29 | 25 | 62 | 5 |
| 5 | | Trimethyl orthoacetate (**Alk-D1**) | 7 | 42 | 44 | 60 | 6 |
| 6 | | Triethyl orthoacetate (**Alk-D2**) | 24 | 41 | 50 | 71 | 10 |
| 7 | | Trimethyl orthobenzoate (**Alk-D3**) | 36 | 38 | 27 | 44 | 3 |
| 8 | *rac*-**2** | Triethyl orthobenzoate (**Alk-D4**) | 7 | 38 | 27 | 45 | 3 |

[a] Conditions: *rac*-**1**–**2** 87 μmol, Novozym 435-STREM 10 mg, PhCH$_3$ 2 mL, alkoxy group donor (**Alk-D1–D4**, 3 equiv), 40 °C, 800 rpm (magnetic stirrer); [b] Calculated from the enantiomeric excess of the unreacted carboxylic acid (ee$_s$) and the formed ester (ee$_p$) according to the formula conv. = ee$_s$/(ee$_s$ + ee$_p$); [c] Determined by chiral HPLC analysis using a (*S*,*S*)-Whelk-O 1 column; [d] Calculated according to Chen et al. [65], using the equation: $E = \{\ln[(1 - \text{conv.})(1 - \text{ee}_s)]\}/\{\ln[(1 - \text{conv.})(1 + \text{ee}_s)]\}$.

In a model EKR of *rac*-**1** and *rac*-**2** using various orthoesters **Alk-D1–D4**, Novozym 435-STREM was chosen as the biocatalysts. All reactions were carried out, with 3 equiv of the corresponding **Alk-D1–D4** in PhCH$_3$ at 40 °C for 7–48 h. The results show that trimethyl orthoacetate (**Alk-D1**) was optimal for the resolution of *rac*-**1**, yielding slower-reacting enantiomer (*S*)-(+)-**1** in 55% ee and the formed ester (*R*)-(−)-**1a** in 65% ee, with a reasonably high 46% conv.

When considering the resolution of *rac*-**2**, slightly better results in terms of enantioselectivity were obtained with triethyl orthoacetate (**Alk-D2**) (*E* = 10), which allowed us to achieve 41% conv., yielding enantiomerically enriched acid (*S*)-(+)-**2** with 50% ee and the optically active corresponding ethyl ester (*R*)-(−)-**2b** with 71% ee, respectively. Based on these findings, it was somehow puzzling why **Alk-D2** turned out to be utterly inefficient in the case of naproxen (*rac*-**1**). The same phenomenon was observed in the series of trialkyl orthobenzoates. The ethyl esters of these reagents were significantly less efficient in the

reactions with *rac*-**1** than their methyl counterparts. In general, it was disappointing that trialkyl orthobenzoates failed to give the desired improvement in enantioselectivity of biocatalytic reactions, which is in sharp contrast to the results obtained by Ostaszewski and co-workers during EKR of 3-phenyl-4-pentenoic acid [60]. It is noteworthy that a switch in stereopreference of CAL-B in the presence of tested orthoesters **Alk-D1–D4** was not observed.

Conducting the lipase-catalyzed reactions in a non-aqueous environment is beneficial for synthetic purposes, mainly because the improved solubility of the hydrophobic compounds in organic solvents extends the potential substrate scope for such biotransformations. Moreover, it is well documented that the organic solvent strongly influences biocatalytic reactions and may tremendously potentiate the lipases' catalytic performances. This is mainly due to the properties of organic media that make a protein structure more rigid and thus more stable against thermal denaturation [67]. According to the literature, the organic solvents with a log *p* value > 2 are recommended for lipase-catalyzed reactions [68]. In general, the correlation between polarity and activity of lipases in non-aqueous media parallels the ability of organic solvents to distort the essential water layer that stabilizes the biocatalyst. Thus, the water-miscible compounds possessing the water-stripping ability are harmful to enzymes' stability and activity. There is also clear empirical evidence that the polarity of the solvents may also play a critical role in penetration and occupation of the enzyme active site [69,70] as well as alteration of the solvation of the transition state [71–73] by the organic solvent. In rare cases, the organic solvents can also alter the stereopreference of lipases toward racemates [74] and/or their regioselectivity of action [75].

In order to determine the optimal medium for the kinetic resolution of the selected racemic profens *rac*-**1** and *rac*-**2**, a set of different organic solvents with varying log *p* values (from −0.31 to 3.75) were studied under similar conditions (Table 3). In general, these reactions were catalyzed by Novozym 435-STREM and conducted in an organic solvent at 40 °C for 72 h (in the case of *rac*-**1**) or 7 h (in the case of *rac*-**2**), respectively. Furthermore, trimethyl orthoacetate (**Alk-D1**) was used as a donor of the alkoxy group in the case of *rac*-**1**, whereas for the EKR of *rac*-**2**, triethyl orthoacetate (**Alk-D2**) was employed. The results of Novozym 435-STREM-catalyzed EKR of both profens *rac*-**1** and *rac*-**2** revealed that most of the polar aprotic solvents (i.e., 1,4-dioxane, $CH_3CN$, acetone, THF), as well as semipolar 2-methyl-2-butanol (*tert*-amyl alcohol) and $CHCl_3$, were detrimental for the catalytic activity of CAL-B. Interestingly, the only solvent with a log *p* value below 2 in which the suspended CAL-B could catalyze the reactions, was $CH_2Cl_2$. This result suggests that the solvents' polarity was probably not a decisive factor in this case. There might also be other specific interactions that could affect the catalytic behavior of CAL-B, such as additional halogen bonding formed between chlorinated solvent and the biocatalyst. It has already been proven that halogen interactions in protein–ligand complexes can modulate the peptide/protein conformations and significantly change their catalytic properties [76–78]. Nevertheless, for both studied substrates, the biotransformations carried out in $CH_2Cl_2$ reached less than 30% conv., and gave only *E* = 1. In turn, the employment of hydrophobic solvents characterized by log *p* > 2.5 turned out to be more compatible with the biocatalytic system and thus significantly improved the rates of CAL-B-catalyzed KR of *rac*-**1** and *rac*-**2**. It clearly indicated that CAL-B in hydrophobic solvents with a high log *p* value shows good activity compared to organic solvents characterized by a low log *p*-value.

In the case of *rac*-**1**, the highest conversions were achieved in cyclohexane (72%) and isooctane (89%), albeit this media negatively affected enantioselectivity (*E* = 1–2). The best catalytic efficiency of Novozym 435-STREM in the kinetic resolution of *rac*-**1** was observed when $PhCH_3$ or xylene was used as the respective reaction media. Only these solvents favored the transformation of *rac*-**1** with *E* = 4–8, yielding enantiomerically enriched acid (*S*)-(+)-**1** in a 55–61% ee range and ester (*R*)-(+)-**1a** in a 38–65% ee range with 46–62% conv. (Table 3, entries 9 and 11), respectively. Although xylene gave better results in terms of the % ee-values of the slower-reacting (*S*)-enantiomer, $PhCH_3$ is more volatile and easier

to remove from the reaction mixture by evaporation. Therefore, further EKR experiments with *rac*-**1** were carried out by using PhCH₃ as a solvent.

**Table 3.** Co-solvent screening for (Novozym 435-STREM)-catalyzed KR of naproxen (*rac*-**1**) with **Alk-D1** at 40 °C for 72 h and ibuprofen (*rac*-**2**) with **Alk-D2** at 40 °C for 7 h.

| Entry | Substrate [a] | Solvent (log *p*) [b] | Time (h) | Conv. (%) [c] | ee$_s$ (%) [d] | ee$_p$ (%) [d] | *E* [e] |
|---|---|---|---|---|---|---|---|
| 1 | | 1,4-Dioxane (−0.31) | 72 | 0 | N.D. [f] | N.D. [f] | N.D. [f] |
| 2 | | CH₃CN (0.17) | 72 | 0 | N.D. [f] | N.D. [f] | N.D. [f] |
| 3 | | Acetone (0.20) | 72 | 0 | N.D. [f] | N.D. [f] | N.D. [f] |
| 4 | | THF (0.40) | 72 | 0 | N.D. [f] | N.D. [f] | N.D. [f] |
| 5 | | CH₂Cl₂ (1.01) | 72 | 27 | 3 | 8 | 1 |
| 6 | | *t*-Amyl alcohol (1.09) | 72 | 0 | N.D. [f] | N.D. [f] | N.D. [f] |
| 7 | | CHCl₃ (1.67) | 72 | 0 | N.D. [f] | N.D. [f] | N.D. [f] |
| 8 | *rac*-**1** | Cyclohexane (2.50) | 72 | 72 | 26 | 10 | 2 |
| 9 | | PhCH₃ (2.52) | 72 | 46 | 55 | 65 | 8 |
| 10 | | *n*-Hexane (3.00) | 72 | 64 | 16 | 9 | 1 |
| 11 | | Xylene (3.01) | 72 | 62 | 61 | 38 | 4 |
| 12 | | Isooctane (3.75) | 72 | 89 | 23 | 3 | 1 |
| 13 | | 1,4-Dioxane (−0.31) | 7 | 0 | N.D. [f] | N.D. [f] | N.D. [f] |
| 14 | | CH₃CN (0.17) | 7 | 0 | N.D. [f] | N.D. [f] | N.D. [f] |
| 15 | | Acetone (0.20) | 7 | 0 | N.D. [f] | N.D. [f] | N.D. [f] |
| 16 | | THF (0.40) | 7 | 0 | N.D. [f] | N.D. [f] | N.D. [f] |
| 17 | | CH₂Cl₂ (1.01) | 7 | 17 | 2 | 10 | 1 |
| 18 | | *t*-Amyl alcohol (1.09) | 7 | 0 | N.D. [f] | N.D. [f] | N.D. [f] |
| 19 | | CHCl₃ (1.67) | 7 | 0 | N.D. [f] | N.D. [f] | N.D. [f] |
| 20 | *rac*-**2** | Cyclohexane (2.50) | 7 | 47 | 54 | 62 | 7 |
| 21 | | PhCH₃ (2.52) | 7 | 19 | 17 | 75 | 8 |
| 22 | | *n*-Hexane (3.00) | 7 | 59 | 74 | 52 | 7 |
| 23 | | Xylene (3.01) | 7 | 37 | 29 | 49 | 4 |
| 24 | | Isooctane (3.75) | 7 | 57 | 63 | 48 | 5 |

[a] Conditions: *rac*-**1**–**2** 87 µmol, Novozym 435-STREM 10 mg, organic solvent 2 mL, 3 equiv of trimethyl orthoacetate (**Alk-D1** for *rac*-**1**) or 3 equiv of triethyl orthoacetate (**Alk-D2** for *rac*-**2**), 40 °C, 800 rpm (magnetic stirrer); [b] Logarithm of the partition coefficient of a given solvent between *n*-octanol and water according to ChemBioDraw Ultra 13.0 software indications; [c] Calculated from the enantiomeric excess of the unreacted carboxylic acid (ee$_s$) and the formed ester (ee$_p$) according to the formula conv. = ee$_s$/(ee$_s$ + ee$_p$); [d] Determined by chiral HPLC analysis using a (*S,S*)-Whelk-O 1 column; [e] Calculated according to Chen et al. [65], using the equation: $E = \{\ln[(1 - \text{conv.})(1 - \text{ee}_s)]\}/\{\ln[(1 - \text{conv.})(1 + \text{ee}_s)]\}$; [f] Not determined.

On the other hand, the results of the reactions conducted with *rac*-**2** showed that *n*-hexane was the solvent of choice as it boosted the rate of the reaction 3-fold with respect to the EKR carried out in PhCH₃ and allowed us to obtain the desired enantiomer (*S*)-(+)-**2** with the highest being 74% ee and 59% conv. after 7 h.

In the next step of our studies, we have extended the scope of the potential racemic substrates of another three profens, namely ketoprofen (*rac*-**3**), flurbiprofen (*rac*-**4**) and etodolac (*rac*-**5**), respectively (Table 4). In particular, etodolac seemed challenging since, until now, only two literature reports have presented the EKR of its racemic mixture [79,80]; however, poor conversions and enantioselectivities were observed. Enzymatic resolutions of the above-mentioned NSAIDs were performed using the hitherto optimized conditions with an arbitrary imposed alkoxy group reagent. For all of the tested substrates, triethyl orthoacetate (**Alk-D2**) was employed as an alkoxy donor. Owing to the promising results

of the enantioselective esterification of *rac-***1** and *rac-***2** with orthoesters in hydrophobic solvents, the effect of both *n*-hexane and PhCH$_3$ was evaluated for each substrate after an arbitrary set 48 h.

**Table 4.** Novozym 435-STREM-catalyzed KR of ketoprofen (*rac-***3**), flurbiprofen (*rac-***4**), and etodolac (*rac-***5**) with **Alk-D2** in *n*-hexane or PhCH$_3$ at 40 °C for 48 h.

| Entry | Substrate [a] | Solvent | Conv. (%) [b] | ee$_s$ (%) [c] | ee$_p$ (%) [c] | E [d] |
|---|---|---|---|---|---|---|
| 1 | *rac-***3** | *n*-Hexane | 77 | 69 | 21 | 3 |
| 2 | | PhCH$_3$ | 16 | 4 | 21 | 2 |
| 3 | *rac-***4** | *n*-Hexane | 66 | 97 | 49 | 11 |
| 4 | | PhCH$_3$ | 53 | 81 | 71 | 14 |
| 5 | *rac-***5** | *n*-Hexane | 0 | N.D. [e] | N.D. [e] | N.D. [e] |
| 6 | | PhCH$_3$ | 0 | N.D. [e] | N.D. [e] | N.D. [e] |

[a] Conditions: *rac-***3**–**4** 87 μmmol, Novozym 435-STREM 10 mg, organic solvent 2 mL, triethyl orthoacetate (**Alk-D2**, 3 equiv), 40 °C, 800 rpm (magnetic stirrer); [b] Calculated from the enantiomeric excess of the unreacted carboxylic acid (ee$_s$) and the formed ester (ee$_p$) according to the formula conv. = ee$_s$/(ee$_s$ + ee$_p$); [c] Determined by chiral HPLC analysis by using a Chiralpak OJ-H column; [d] Calculated according to Chen et al. [65], using the equation: $E = \{\ln[(1 - \text{conv.})(1 - \text{ee}_s)]\}/\{\ln[(1 - \text{conv.})(1 + \text{ee}_s)]\}$; [e] Not determined.

The results summarized in Table 4 indicate that the best analytical-scale lipase-catalyzed KRs of *rac-***3** and *rac-***4** in terms of enantioselectivity, the optical purity of the desired products (*S*)-(+)-**3** and (*S*)-(+)-**4**, and reaction rate were obtained by employing *n*-hexane as the medium. In the case of EKR of ketoprofen (*rac-***3**), the remaining acid (*S*)-(+)-**3** was isolated with 69% ee; whereas the EKR of flurbiprofen (*rac-***4**) resulted in the isolation of the slower-reacting enantiomer (*S*)-(+)-**4** with 97% ee. The stereochemical outcome of the EKR of *rac-***4** is in line with the results reported by Morrone et al. [59]

In addition, the differences in the reaction rates noticed for both solvents used in the case of *rac-***3** were alike for racemic ibuprofen (*rac-***2**). This phenomenon was not observed for the EKR of *rac-***4**, in which both solvents gave similar conversions and *E*-values. To our great disappointment, Novozym 435-STREM failed to catalyze the KR of racemic etodolac (*rac-***5**) (Table 4, entries 5 and 6). It is clear that this result might be attributed to the structure of the substrate *rac-***5**, which is too bulky to be accepted by the CAL-B active site.

Furthermore, the usefulness of the developed biocatalytic method was demonstrated by the investigation into the 0.43 mmol scale enantioselective esterification of racemic NSAIDs *rac-***1**–**4** (Table 5). The reaction times and the type of trialkyl orthoacetate (i.e., **Alk-D1**, **Alk-D2**) were adjusted independently for each substrate *rac-***1**–**4**, according to the results obtained during the optimization studies (see above). Satisfyingly, preparative-scale lipase-catalyzed EKR of *rac-***1**–**4** allowed us to achieve a relatively high 46–73% conv. after 7–72 h with optically active profens (*S*)-(+)-**1**–**4**, isolated in a 23–42% yield range and with 56–97% ee, respectively. In turn, the corresponding (*R*)-esters, including (*R*)-(−)-**1a** and (*R*)-(−)-**2**–**4b** were obtained in a 32–52% yield range and with 25–67% ee, respectively. The method was not further optimized; however, it is evident that elongation of the reaction time, i.e., for ibuprofen (*rac-***2**), should allow us to obtain (*S*)-(+)-**2** with higher conversions and an improved optical purity.

**Table 5.** Preparative-scale (Novozym 435-STREM)-catalyzed KR of racemic profens *rac*-**1**–**4** using trialkyl orthoesters **Alk-D1**–**D2** as alkoxy donors.

| Entry | Substrate [a] | Alkoxy Donor | Solvent | Time (h) | Conv. (%) [b] | ees (%) [c]/ Yield (%) [d] | eep (%) [c]/ Yield (%) [d] | E [e] |
|---|---|---|---|---|---|---|---|---|
| 1 | *rac*-**1** | **Alk-D1** | PhCH₃ | 72 | 66 | 57/42 | 29/49 | 3 |
| 2 | *rac*-**2** | **Alk-D2** | *n*-Hexane | 7 | 46 | 56/45 | 67/32 | 9 |
| 3 | *rac*-**3** | **Alk-D2** | *n*-Hexane | 48 | 73 | 69/23 | 25/45 | 3 |
| 4 | *rac*-**4** | **Alk-D2** | PhCH₃ | 48 | 66 | 97/28 | 51/52 | 12 |

[a] Conditions: *rac*-**1**–**4** 0.43 mmol, Novozym 435-STREM 50 mg, organic solvent 10 mL, trialkyl orthoacetate (**Alk-D1**–**D2**, 3 equiv), 40 °C, 800 rpm (magnetic stirrer); [b] Calculated from the enantiomeric excess of the unreacted carboxylic acid ($ee_s$) and the formed ester ($ee_p$) according to the formula conv. = $ee_s/(ee_s + ee_p)$; [c] Determined by chiral HPLC analysis by using analytical columns packed with various CSPs (see Supporting Information); [d] Isolated yield after column chromatography eluted sequentially with *n*-hexane/AcOEt (70:30, *v/v*) and CHCl₃/MeOH (50:50, *v/v*) mixtures; [e] Calculated according to Chen et al. [65], using the equation: $E = \{\ln[(1 - \text{conv.})(1 - ee_s)]\}/\{\ln[(1 - \text{conv.})(1 + ee_s)]\}$.

## 3. Materials and Methods

All non-steroidal anti-inflammatory drugs (NSAIDs), as well as trialkyl orthoesters, were purchased from TCI—Tokyo Chemical Industry. HPLC-grade acetic acid glacial (Catalog No. A35-500) and trifluoroacetic acid (Catalog No. 02-004-498) were purchased from Fisher Scientific. HPLC grade *n*-hexane and 2-propanol (IPA) were purchased from Avantor Performance Materials Poland S.A. (formerly POCH Polish Chemicals Reagents). The enzyme preparations were purchased from Novozymes A/S (Bagsvaerd, Denmark), Roche (Basel, Switzerland), Sigma Aldrich (currently Merck) (Darmstadt, Germany), STREM Chemicals, Inc. (Newburyport, MA, USA), Boehringer Mannheim (currently Roche Diagnostics) (Basel, Switzerland), and Amano Pharmaceutical Co., Ltd. (Nagoya, Japan), and were used without pre-treatment (for details see Table S1 appended in Supporting Information).

Melting points, uncorrected, were determined with a commercial apparatus (Thomas-Hoover "UNI-MELT" capillary melting point apparatus from Arthur H. Thomas Co., (Philadelphia, PA, USA)) on samples contained in rotating capillary glass tubes open on one side (1.35 mm inner diam. and 80 mm length).

Analytical thin-layer chromatography was carried out on TLC aluminum plates (Merck) covered with a silica gel of 0.2 mm thickness film containing a fluorescence indicator green 254 nm (F254) and UV light as a visualizing agent.

The chromatographic analyses (GC) were performed with an Agilent Technologies 6890N instrument, from Agilent Technologies, Inc. (Santa Clara, CA, USA), equipped with a flame ionization detector (FID) and fitted with a HP-50+ (30 m) semipolar column (50%

phenyl–50% methylpolysiloxane); Helium (2 mL/min) was used as a carrier gas; retention times ($t_R$) are given in minutes under these conditions.

The enantiomeric excesses (% ee) of kinetic resolution products were determined by HPLC analysis performed on Shimadzu Nexera-*i* (LC-2040C 3D), from Shimadzu Corp. (Kioto, Japan), equipped with a photodiode array detector (PAD) using columns packed with chiral stationary phases as follows: Chiralcel OJ-H (4.6 mm × 250 mm, coated on 5 μm silica gel grain size, from Daicel Chemical Ind., Ltd. (Minato City, Tokyo, Japan)) or Chiralpak AD-H (4.6 mm × 250 mm, coated on 5 μm silica gel grain size, from Daicel Chemical Ind., Ltd.) or (*S*,*S*)-Whelk-O 1 (4.6 mm × 250 mm, coated on 5 μm silica gel grain size, from Regis Pirkle Technologies, Inc. (Morton Grove, IL, USA)) all of them were equipped with a pre-column (4 mm × 10 mm, 5 μm) using mixtures of *n*-hexane/2-PrOH as a mobile phase without or with an additive (acetic acid (AA) or trifluoroacetic acid (TFA)) in the appropriate ratios given in the experimental section (both the mobile phase composition as well as the flow rate were fine-tuned for each analysis (see Table S2 appended in Supporting Information)); the HPLC analyses were executed in an isocratic and isothermal (30 °C) manner.

Optical rotations ($[\alpha]$) were measured with a PolAAr 32 polarimeter, from Optical Activity Ltd. (Huntingdon, UK), in a 2 dm long cuvette using the sodium D line ($\lambda$ = 589 nm).

$^1$H NMR (500 MHz) and $^{13}$C NMR (126 MHz) spectra were recorded on a Varian NMR System 500 MHz spectrometer, from Varian, Inc. (Palo Alto, CA, USA); $^1$H, $^{13}$C and $^{19}$F chemical shifts ($\delta$) are reported in parts per million (ppm) relative to the solvent signals {CDCl$_3$, $\delta_H$ (residual CHCl$_3$) 7.26 ppm, $\delta_C$ 77.16 ppm or DMSO-*d$_6$*: $\delta_H$ [residual (CD$_3$)$_2$SO] = 2.49 ppm with HDO at $\delta_H$ = 3.30 ppm, $\delta_C$ = 40.45 ppm} or internal CFCl$_3$ reference set at 0 ppm; all of the raw $^1$H and $^{13}$C NMR spectra were created by a non-commercial (freeware) ACD/NMR Processor Academic Edition 12.0 from Advanced Chemistry Development, Inc. ACD/Labs (Toronto, ON, Canada). Chemical shifts are quoted as: s (singlet), d (doublet), dd (doublet of doublets), t (triplet), q (quartet), m (multiplet), and br s (broad singlet); coupling constants (*J*) are reported in Hertz (Hz).

Mass spectrometry was recorded on Micro-mass ESI Q-TOF spectrometer with MSI concept 1H (EI, 70 eV ionization) for MS analysis and on Q Exactive Hybrid Quadrupole-Orbitrap Mass Spectrometer both from Waters Corporation (Milford, MA, USA), ESI source: electrospray with spray voltage 4.00 kV for FTMS analysis; all samples were prepared by dilution of MeOH (0.5 mL) and additives of mixtures of CH$_3$CN/MeOH/H$_2$O (50:25:25, *v/v/v*) + 0.5% formic acid (HCO$_2$H) each.

IR spectra were recorded on Specord M80 from Carl Zeiss (Jena, Germany) in transmittance mode in the 300–4000 cm$^{-1}$ range, in ambient air at room temperature, with 2 cm$^{-1}$ resolution and accumulation of 32 scans; wavenumber (frequency, $\nu$) is given in cm$^{-1}$; samples were prepared as Nujol suspensions.

### 3.1. General Procedure for the Synthesis of Racemic NSAIDs' Esters rac-**1a**–**b**, rac-**2a**–**b**, rac-**3a**–**b**, rac-**4a**–**b**, rac-**5a**–**b**

To a mixture of the respective NSAID (*rac*-**1–5**, 1.05 mmol), MeOH (37 mg, 1.16 mmol, 47 μL) or EtOH (53 mg, 1.16 mmol, 68 μL) and DMAP (57 mg, 0.46 mmol) in CH$_2$Cl$_2$ (10 mL), EDCI hydrochloride (242 mg, 1.26 mmol) was added in one portion at 0–5 °C. Next, the reaction mixture was slowly warmed to room temperature and stirred for 12 h. After this, the content of the flask was diluted with CH$_2$Cl$_2$ (5 mL), washed with H$_2$O (4 × 15 mL), and the aqueous layer was extracted with CH$_2$Cl$_2$ (2 × 25 mL). The combined organic phases were quenched with H$_2$O (25 mL) and brine (25 mL), dried over anhydrous Na$_2$SO$_4$, filtered, and the permeate was concentrated in a vacuum. The crude residue was purified by silica gel chromatography using a mixture of hexane/AcOEt (70:30, *v/v*) to provide the corresponding ester of NSAID *rac*-**1a**–**b**, *rac*-**2a**–**b**, *rac*-**3a**–**b**, *rac*-**4a**–**b**, *rac*-**5a**–**b**.

Methyl 2-(6-methoxynaphthalen-2-yl)propanoate (naproxen methyl ester, *rac*-**1a**). Yield 84% (224 mg); white solid; mp 70–71 °C (CH$_2$Cl$_2$) [lit. [81] 70–72 °C (Petroleum ether/AcOEt)]; $R_f$ [hexane/AcOEt (70:30, *v/v*)] 0.84; $^1$H NMR (500 MHz, CDCl$_3$): δ 1.59 (d,

$J$ = 7.1 Hz, 3H), 3.66–3.69 (m, 3H), 3.87 (q, $J$ = 7.2 Hz, 1H), 3.92 (s, 3H), 7.10–7.17 (m, 2H), 7.38–7.44 (m, 1H), 7.66–7.68 (m, 1H), 7.69–7.74 (m, 2H); $^{13}$C NMR (126 MHz, CDCl$_3$): $\delta$ 18.7, 45.5, 52.2, 55.4, 105.7, 119.1, 126.1, 126.3, 127.3, 129.1, 129.4, 133.8, 135.8, 157.8, 175.3; IR (nujol): $\nu_{max}$ = 2932, 1740, 1604, 1460, 1376, 1268, 1176, 1028, 924, 896, 856, 824; MS (ESI-TOF) $m/z$: [M + H]$^+$ Calcd for C$_{15}$H$_{17}$O$_3^+$ $m/z$: 245.1172, Found 245.1324; FTMS (ESI-TOF) $m/z$: [M + H]$^+$ Calcd for C$_{15}$H$_{17}$O$_3^+$ $m/z$: 245.11722, Found 245.11714; HPLC [*n*-hexane-2-PrOH (70:30, $v/v$); f = 0.8 mL/min; $\lambda$ = 230 nm; $T$ = 30 °C, $p$ = 3.9 MPa, (*S,S*)-Whelk-O 1]: $t_R$ = 9.797 (*R*-isomer) and 12.541 min (*S*-isomer); GC [220–260 (10 °C/min)]: $t_R$ = 3.93 min.

Ethyl 2-(6-methoxynaphthalen-2-yl)propanoate (naproxen ethyl ester, *rac*-**1b**). Yield 86% (241 mg); white solid; mp 62–63 °C (CH$_2$Cl$_2$); $R_f$ [hexane/AcOEt (70:30, $v/v$)] 0.76; $^1$H NMR (500 MHz, CDCl$_3$): $\delta$ 1.21 (t, $J$ = 7.1 Hz, 3H), 1.58 (d, $J$ = 7.1 Hz, 3H), 3.84 (q, $J$ = 7.1 Hz, 1H), 3.92 (s, 3H), 4.07–4.20 (m, 2H), 7.10–7.17 (m, 2H), 7.40–7.44 (m, 1H), 7.66–7.74 (m, 3H); $^{13}$C NMR (126 MHz, CDCl$_3$): $\delta$ 14.3, 18.8, 45.6, 55.4, 60.9, 105.7, 119.1, 126.0, 126.4, 127.2, 129.1, 129.4, 133.8, 136.0, 157.7, 174.8; IR (nujol): $\nu_{max}$ = 2924, 1732, 1604, 1460, 1376, 1264, 1180, 1028, 856, 824; MS (ESI-TOF) $m/z$: [M + H]$^+$ Calcd for C$_{16}$H$_{19}$O$_3^+$ $m/z$: 259.1329, Found 259.1118; FTMS (ESI-TOF) $m/z$: [M + H]$^+$ Calcd for C$_{16}$H$_{19}$O$_3^+$ $m/z$: 259.13287, Found 259.13280; HPLC [*n*-hexane-2-PrOH (70:30, $v/v$); f = 0.8 mL/min; $\lambda$ = 230 nm; $T$ = 30 °C, $p$ = 3.9 MPa, (*S,S*)-Whelk-O 1]: $t_R$ = 8.975 (*R*-isomer) and 11.542 min (*S*-isomer); GC [220–260 (10 °C/min)]: $t_R$ = 4.15 min.

Methyl 2-[4-(2-methylpropyl)phenyl]propanoate (ibuprofen methyl ester, *rac*-**2a**). Yield 75% (173 mg); colourless oil; $R_f$ [hexane/AcOEt (70:30, $v/v$)] 0.91; $^1$H NMR (500 MHz, CDCl$_3$): $\delta$ 0.90 (d, $J$ = 6.6 Hz, 6H), 1.49 (d, $J$ = 7.1 Hz, 3H), 1.80–1.91 (m, 1H), 2.45 (d, $J$ = 7.1 Hz, 2H), 3.66 (s, 3H), 3.70 (d, $J$ = 7.3 Hz, 1H), 7.07–7.12 (m, 2H), 7.18–7.22 (m, 2H); $^{13}$C NMR (126 MHz, CDCl$_3$): $\delta$ 18.8, 22.5, 30.3, 45.2, 45.2, 52.1, 127.3, 129.5, 137.9, 140.7, 175.4; IR (nujol): $\nu_{max}$ = 2956, 1740, 1460, 1164; MS (ESI-TOF) $m/z$: [M + H]$^+$ Calcd for C$_{14}$H$_{21}$O$_2^+$ $m/z$: 221.1536, Found 221.1591; FTMS (ESI-TOF) $m/z$: [M + H]$^+$ Calcd for C$_{14}$H$_{21}$O$_2^+$ $m/z$: 221.15361, Found 221.15350; HPLC [*n*-hexane-2-PrOH (99:1, $v/v$); f = 0.7 mL/min; $\lambda$ = 217 nm; $T$ = 30 °C, $p$ = 2.8 MPa, Chiralcel OJ-H]: $t_R$ = 10.393 (*S*-isomer) and 13.210 min (*R*-isomer); GC [150–260 (10 °C/min)]: $t_R$ = 3.86 min.

Ethyl 2-[4-(2-methylpropyl)phenyl]propanoate (ibuprofen ethyl ester, *rac*-**2b**). Yield 57% (163 mg); colourless oil; hexane/AcOEt (70:30, $v/v$)] 0.80; $^1$H NMR (500 MHz, CDCl$_3$): $\delta$ 0.90 (d, $J$ = 6.6 Hz, 6H), 1.21 (t, $J$ = 7.1 Hz, 3H), 1.48 (d, $J$ = 7.1 Hz, 3H), 1.79–1.90 (m, 1H), 2.45 (d, $J$ = 7.1 Hz, 2H), 3.68 (q, $J$ = 7.1 Hz, 1H), 4.05–4.19 (m, 2H), 7.07–7.12 (m, 2H), 7.18–7.23 (m, 2H); $^{13}$C NMR (126 MHz, CDCl$_3$): $\delta$ 14.3, 18.8, 22.5, 30.3, 45.2, 45.3, 60.8, 127.3, 129.26, 138.0, 140.6, 174.9; IR (nujol): $\nu_{max}$ = 2956, 1736, 1164; MS (ESI-TOF) $m/z$: [M + H]$^+$ Calcd for C$_{15}$H$_{23}$O$_2^+$ $m/z$: 235.1693, Found 235.1691; FTMS (ESI-TOF) $m/z$: [M + H]$^+$ Calcd for C$_{15}$H$_{23}$O$_2^+$ $m/z$: 235.16926, Found 235.16913; HPLC [*n*-hexane-2-PrOH (99:1, $v/v$); f = 0.7 mL/min; $\lambda$ = 217 nm; $T$ = 30 °C, $p$ = 2.8 MPa, Chiralcel OJ-H]: $t_R$ = 8.465 (*S*-isomer) and 10.159 min (*R*-isomer); GC [150–260 (10 °C/min)]: $t_R$ = 4.20 min.

Methyl 2-(3-benzoylphenyl)propanoate (ketoprofen methyl ester, *rac*-**3a**). Yield 82% (230 mg); colourless oil; $R_f$ [hexane/AcOEt (70:30, $v/v$)] 0.69; $^1$H NMR (500 MHz, CDCl$_3$): $\delta$ 1.54 (d, $J$ = 7.3 Hz, 3H), 3.67 (s, 3H), 3.81 (q, $J$ = 7.2 Hz, 1H), 7.41–7.51 (m, 3H), 7.52–7.56 (m, 1H), 7.57–7.62 (m, 1H), 7.65–7.70 (m, 1H), 7.73–7.84 (m, 3H); $^{13}$C NMR (126 MHz, CDCl$_3$): $\delta$ 18.6, 45.4, 52.3, 128.4, 128.7, 129.1, 129.3, 130.2, 131.6, 132.6, 137.6, 138.0, 141.0, 174.6, 196.6; IR (nujol): $\nu_{max}$ = 2952, 1740, 1660, 1596, 1448, 1376, 1284, 1208, 1076, 1000, 952, 860, 820, 724, 644; MS (ESI-TOF) $m/z$: [M + H]$^+$ Calcd for C$_{17}$H$_{17}$O$_3^+$ $m/z$: 269.1172, Found 269.0729; FTMS (ESI-TOF) $m/z$: [M + H]$^+$ Calcd for C$_{17}$H$_{17}$O$_3^+$ $m/z$: 269.11722, Found 269.11710; HPLC [*n*-hexane-2-PrOH (99:1, $v/v$); f = 0.7 mL/min; $\lambda$ = 249 nm; $T$ = 30 °C, $p$ = 2.8 MPa, Chiralpak AD-H]: $t_R$ = 29.146 (*S*-isomer) and 30.214 min (*R*-isomer).

Ethyl 2-(3-benzoylphenyl)propanoate (ketoprofen ethyl ester, *rac*-**3b**). Yield 71% (209 mg); colourless oil; $R_f$ [hexane/AcOEt (70:30, $v/v$)] 0.72; $^1$H NMR (500 MHz, CDCl$_3$): $\delta$ 1.22 (t, $J$ = 7.3 Hz, 3H), 1.53 (d, $J$ = 7.3 Hz, 3H), 3.78 (q, $J$ = 7.3 Hz, 1H), 4.07–4.20 (m, 2H), 7.41–7.51 (m, 3H), 7.53–7.62 (m, 2H), 7.66–7.70 (m, 1H) 7.75 (t, $J$ = 1.7 Hz, 1H) 7.78–7.82 (m, 2H); $^{13}$C NMR (126 MHz, CDCl$_3$): $\delta$ 14.3, 18.6, 45.6, 61.1, 128.4, 128.7, 129.1, 129.4, 130.2,

131.6, 132.6, 137.7, 138.0, 141.1, 174.2, 196.6; IR (nujol): $\nu_{max}$ = 2984, 1736, 1656, 1448, 1176, 720; MS (ESI-TOF) *m/z*: $[M + H]^+$ Calcd for $C_{18}H_{19}O_3^+$ *m/z*: 283.1329, Found 283.1163; FTMS (ESI-TOF) *m/z*: $[M + H]^+$ Calcd for $C_{18}H_{19}O_3^+$ *m/z*: 283.13287, Found 283.13283; HPLC [*n*-hexane-2-PrOH (99:1, *v/v*); f = 0.7 mL/min; λ = 249 nm; *T* = 30 °C, *p* = 2.8 MPa, Chiralcel OJ-H]: $t_R$ = 30.411 (*S*-isomer) and 31.951 min (*R*-isomer).

Methyl 2-(2-fluoro [1,1′-biphenyl]-4-yl)propanoate (flurbiprofen methyl ester, *rac-***4a**). Yield 83% (225 mg); white solid; mp 48–50 °C (hexane/AcOEt) [lit. [82] 45–46 °C (AcOEt/petrol ether)]; $R_f$ [hexane/AcOEt (70:30, *v/v*)] 0.69; [1]H NMR (500 MHz, CDCl₃): δ 1.55 (d, *J* = 6.9 Hz, 3H), 3.71 (s, 3H), 3.77 (q, *J* = 7.3 Hz, 1H), 7.10–7.18 (m, 2H), 7.34–7.48 (m, 4H), 7.52–7.57 (m, 2H); [13]C NMR (126 MHz, CDCl₃): δ 18.6, 45.1, 52.4, 115.4 (d, $J_{C-F}$ = 3.9 Hz), 123.65 (d, $J_{C-F}$ = 3.9 Hz), 127.8, 128.0 (d, $J_{C-F}$ = 13.7 Hz), 128.6, 129.1 (d, $J_{C-F}$ = 2.9 Hz), 131.0 (d, $J_{C-F}$ = 3.9 Hz), 135.6, 141.9 (d, $J_{C-F}$ = 6.9 Hz), 159.8 (d, $J_{C-F}$ = 248.8 Hz), 174.57; [19]F NMR (470 MHz, CDCl₃): δ −117.6; IR (nujol): $\nu_{max}$ = 2936, 1744, 1460, 1376, 1172, 920, 764, 724, 696; MS (ESI-TOF) *m/z*: $[M + H]^+$ Calcd for $C_{16}H_{16}FO_2^+$ *m/z*: 259.1129, Found 259.0921; FTMS (ESI-TOF) *m/z*: $[M + H]^+$ Calcd for $C_{16}H_{16}FO_2^+$ *m/z*: 259.11288, Found 259.11299; HPLC [*n*-hexane-2-PrOH (99:1, *v/v*); f = 0.7 mL/min; λ = 246 nm; *T* = 30 °C, *p* = 2.8 MPa, Chiralcel OJ-H]: $t_R$ = 29.847 (*S*-isomer) and 35.927 min (*R*-isomer).

Ethyl 2-(2-fluoro [1,1′-biphenyl]-4-yl)propanoate (flurbiprofen ethyl ester, *rac-***4b**). Yield 75% (215 mg); colorless oil; $R_f$ [hexane/AcOEt (70:30, *v/v*)] 0.64; [1]H NMR (500 MHz, CDCl₃): δ 1.25 (t, *J* = 7.4 Hz, 3H), 1.54 (d, *J* = 7.3 Hz, 3H), 3.75 (q, *J* = 7.2 Hz, 1H), 4.10–4.23 (m, 2H), 7.10–7.19 (m, 2H), 7.34–7.47 (m, 4H), 7.52–7.57 (m, 2H); [13]C NMR (126 MHz, CDCl₃): δ 14.3, 18.6, 45.2 (d, $J_{C-F}$ = 2.0 Hz), 61.1, 115.4 (d, $J_{C-F}$ = 23.6 Hz), 123.7 (d, $J_{C-F}$ = 2.9 Hz), 127.8, 127.8, 128.6, 129.1 (d, $J_{C-F}$ = 2.9 Hz), 130.9 (d, $J_{C-F}$ = 3.9 Hz), 135.7, 142.1, 159.8 (d, $J_{C-F}$ = 255.0 Hz), 174.1; [19]F NMR (470 MHz, CDCl₃): δ −117.7; IR (nujol): $\nu_{max}$ = 2984, 1732, 1624, 1580, 1560, 1484, 1420, 1376, 1328, 1184, 1132, 1076, 1028, 932, 876, 836, 768, 724, 700; MS (ESI-TOF) *m/z*: $[M + H]^+$ Calcd for $C_{17}H_{18}FO_2^+$ *m/z*: 273.1286, Found 273.1436; FTMS (ESI-TOF) *m/z*: $[M + H]^+$ Calcd for $C_{17}H_{18}FO_2^+$ *m/z*: 273.12853, Found 273.12850; HPLC [*n*-hexane-2-PrOH (99:1, *v/v*); f = 0.7 mL/min; λ = 246 nm; *T* = 30 °C, *p* = 2.8 MPa, Chiralcel OJ-H]: $t_R$ = 19.916 (*S*-isomer) and 26.663 min (*R*-isomer).

Methyl (1,8-diethyl-1,3,4,9-tetrahydropyrano [3,4-b]indol-1-yl)acetate (etodolac methyl ester, *rac-***5a**). Yield 54% (142 mg); white solid; mp 133–134 °C (CHCl₃) [lit. [83] 128–130 °C (EtOH)]; $R_f$ [CHCl₃ (pure)] 0.60; [1]H NMR (500 MHz, DMSO-*d₆*): δ 0.65 (t, *J* = 7.3 Hz, 3H), 1.29 (t, *J* = 7.5 Hz, 3H), 1.93–2.16 (m, 2H), 2.58–2.78 (m, 2H), 2.82–2.94 (m, 3H), 3.05 (d, *J* = 13.7 Hz, 1H), 3.60 (s, 3H), 3.85–3.94 (m, 1H), 3.96–4.05 (m, 1H), 6.88–6.98 (m, 2H), 7.25–7.29 (m, 1H), 10.52 (s, 1H); [13]C NMR (126 MHz, DMSO-*d₆*): δ 8.8, 15.4, 22.8, 24.7, 31.7, 43.4, 52.2, 60.9, 76.3, 108.3, 116.4, 119.7, 120.6, 127.0, 127.5, 135.5, 136.8, 171.0; IR (nujol): $\nu_{max}$ = 3380, 2928, 1708, 1460, 1376, 1316, 1236, 1176, 1080, 744; MS (ESI-TOF) *m/z*: $[M + H]^+$ Calcd for $C_{18}H_{24}NO_3^+$ *m/z*: 302.1751, Found 302.1136; FTMS (ESI-TOF) *m/z*: $[M + H]^+$ Calcd for $C_{18}H_{24}NO_3^+$ *m/z*: 302.17507, Found 302.17476; GC [220–260 (10 °C/min)]: $t_R$ = 5.65 min.

Ethyl (1,8-diethyl-1,3,4,9-tetrahydropyrano [3,4-b]indol-1-yl)acetate (etodolac ethyl ester, *rac-***5b**). Yield 35% (97 mg); white solid; mp 63–66 °C (CHCl₃); $R_f$ [CHCl₃ (pure)] 0.73; [1]H NMR (500 MHz, CDCl₃): δ 0.85 (t, *J* = 7.3 Hz, 3H), 1.23–1.32 (m, 3H), 1.38 (t, *J* = 7.6 Hz, 3H), 2.02 (dq, *J* = 14.6, 7.3 Hz, 1H), 2.18 (dq, *J* = 14.6, 7.4 Hz, 1H), 2.72–2.80 (m, 1H), 2.80–2.95 (m, 4H), 2.99–3.05 (m, 1H), 3.95 (ddd, *J* = 11.5, 7.3, 4.4 Hz, 1H), 4.05 (dt, *J* = 11.4, 4.9 Hz, 1H), 4.12–4.26 (m, 2H), 6.97–7.11 (m, 2H), 7.37 (d, *J* = 7.6 Hz, 1H), 9.12 (br. s., 1H); [13]C NMR (126 MHz, CDCl₃): δ 7.7, 13.9, 14.3, 22.6, 24.4, 30.8, 43.2, 60.8, 61.1, 74.8, 108.5, 116.1, 119.7, 120.5, 126.3, 126.8, 134.6, 136.2, 173.0; IR (nujol): $\nu_{max}$ = 3384, 2920, 1704, 1460, 1376; MS (ESI-TOF) *m/z*: $[M + H]^+$ Calcd for $C_{19}H_{26}NO_3^+$ *m/z*: 316.1907, Found 316.1500; FTMS (ESI-TOF) *m/z*: $[M + H]^+$ Calcd for $C_{19}H_{26}NO_3^+$ *m/z*: 316.19072, Found 316.19061; GC [220–260 (10 °C/min)]: $t_R$ = 5.98 min.

### 3.2. General Procedure for Enzyme Screening for EKR of rac-**1** and rac-**2** through Enantioselective Esterification with Trimethyl Orthoacetate

To a solution, the appropriate racemic profen (i.e., naproxen (*rac*-**1**, 20 mg, 87 μmol) or ibuprofen (*rac*-**2**, 18 mg, 87 μmol)) in PhCH$_3$ (2 mL), trimethyl orthoacetate (**Alk-D1**, 31 mg, 0.26 mmol, 33 μL) and the respective enzyme preparation (10 mg) were added. The reaction mixture was stirred in a thermo-stated screw-capped glass vial ($V$ = 4 mL) placed in an anodized aluminum reaction block at 40 °C and 800 rpm for 72 h (for *rac*-**1**) or 24 h (for *rac*-**2**), respectively. Next, the reaction was stopped by cooling the mixture, filtering off the enzyme on a Schott funnel under vacuum, and washing the enzyme with a portion of PhCH$_3$ (2 mL). After evaporation of the volatiles from the permeate, the resulting crude oil was purified by silica gel column chromatography using subsequent mixtures of hexane/AcOEt (70:30, $v/v$) and CHCl$_3$/MeOH (50:50, $v/v$) as an eluent. In order to obtain data concerning %-conv., % ee as well as *E*-values, the HPLC analyses were performed for the EKR products of each of the employed profen. For HPLC analysis, the representative samples (2–3 mg) were diluted with a mobile phase composed of *n*-hexane/2-PrOH (1.5 mL; 1:1, $v/v$). For additional data, see Table 1 in the main manuscript.

### 3.3. General Procedure for Alkoxy Group Donor Screening for EKR of rac-**1** and rac-**2** through Enantioselective Esterification

To a solution, the appropriate racemic profen (i.e., naproxen (*rac*-**1**, 20 mg, 87 μmol) or ibuprofen (*rac*-**2**, 18 mg, 87 μmol)) in PhCH$_3$ (2 mL), the appropriate alkoxy group donor/trialkyl orthoester (0.26 mmol; for trimethyl orthoacetate (**Alk-D1**): 31 mg, 33 μL); for triethyl orthoacetate (**Alk-D2**): 42 mg, 48 μL; for trimethyl orthobenzoate (**Alk-D3**): 47 mg, 45 μL; for triethyl orthobenzoate (**Alk-D4**): 58 mg, 59 μL) and Novozym 435-STREM (10 mg) were added. The reaction mixture was stirred in a thermo-stated screw-capped glass vial ($V$ = 4 mL) placed in an anodized aluminum reaction block at 40 °C and 800 rpm for 72 h (in the case of *rac*-**1**) and for 7–36 h (in the case of *rac*-**2**). The rest of the manipulations were the same as in the Section 3.2. presented above. For additional data, see Table 2 in the main manuscript.

### 3.4. General Procedure for Organic Solvent Screening for EKR of rac-**1** and rac-**2** through Enantioselective Esterification with Trialkyl Orthoacetates

To a solution of the appropriate racemic profen (i.e., naproxen (*rac*-**1**, 20 mg, 87 μmol) or ibuprofen (*rac*-**2**, 18 mg, 87 μmol)) in the respective organic solvent (2 mL), Novozym 435-STREM (10 mg) and trimethyl orthoacetate (**Alk-D1**, 31 mg, 0.26 mmol, 33 μL; in the case of *rac*-**1**) or triethyl orthoacetate (**Alk-D2**, 42 mg, 0.26 mmol, 48 μL; in the case of *rac*-**2**) were added, respectively. The reaction mixture was stirred in a thermo-stated screw-capped glass vial ($V$ = 4 mL) placed in an anodized aluminum reaction block at 40 °C and 800 rpm for 72 h (for *rac*-**1**) or 7 h (for *rac*-**2**), respectively. The rest of the manipulations were the same as in the above-presented Section 3.2 or Section 3.3. For additional data, see Table 3 in the main manuscript.

### 3.5. General Procedure for EKR of rac-**3**–**5** through Enantioselective Esterification with Triethyl Orthoacetate

To a solution, the appropriate racemic profen (i.e., ketoprofen (*rac*-**3**, 22 mg, 87 μmol) or flurbiprofen (*rac*-**4**, 21 mg, 87 μmol) or etodolac (*rac*-**5**, 25 mg, 87 μmol)) in *n*-hexane or PhCH$_3$ (2 mL), Novozym 435-STREM (10 mg) and triethyl orthoacetate (**Alk-D2**, 42 mg, 0.26 mmol, 48 μL) were added, respectively. The reaction mixture was stirred in a thermo-stated screw-capped glass vial ($V$ = 4 mL) and placed in an anodized aluminum reaction block for 48 h at 40 °C and 800 rpm. The rest of the manipulations were the same as in the above-presented Sections 3.2–3.4. For additional data, see Table 4 in the main manuscript.

*3.6. General Procedure for Preparative Scale EKR of Racemic NSAIDs rac-**1**–**4** through Enantioselective Esterification with Trialkyl Orthoacetates*

To a solution, the appropriate racemic profen (i.e., naproxen (*rac*-**1**, 100 mg, 0.43 mmol) or ibuprofen (*rac*-**2**, 89 mg, 0.43 mmol) or ketoprofen (*rac*-**3**, 109 mg, 0.43 mmol) or flurbiprofen (*rac*-**4**, 105 mg, 0.43 mmol)) in PhCH₃ (10 mL; in the case of *rac*-**1** and *rac*-**4**) or *n*-hexane (10 mL; in the case of *rac*-**2** and *rac*-**3**), trimethyl orthoacetate (**Alk-D1**, 157 mg, 1.30 mmol, 166 μL; in the case of *rac*-**1**) or triethyl orthoacetate (**Alk-D2**, 210 mg, 1.30 mmol, 237 μL; in the case of *rac*-**2**–**4**) and Novozym 435-STREM (50 mg) were added. The reaction mixture was stirred at 40 °C and 800 rpm in the sealed reactor for 72 h (in the case of *rac*-**1**), 7 h (in the case of *rac*-**2**) and 48 h (in the case of *rac*-**3**–**4**), respectively. Next, the reaction was stopped by cooling the mixture, filtering off the enzyme on a Schott funnel under a vacuum, and washing the enzyme with a portion of PhCH₃ (10 mL) or *n*-hexane (10 mL), respectively. After evaporation of the volatiles from the permeate, the resulting crude oil was purified by silica gel column chromatography using subsequent mixtures of *n*-hexane/AcOEt (70:30, *v/v*) and CHCl₃/MeOH (50:50, *v/v*) as an eluent, thus affording enantioenriched profens and their respective esters: (*S*)-naproxen ((*S*)-(+)-**1**, 42 mg, 42% isolated yield, 57% ee) and (*R*)-naproxen methyl ester ((*R*)-(−)-**2a**, 52 mg, 49% isolated yield, 29% ee); (*S*)-ibuprofen ((*S*)-(+)-**2**, 40 mg, 45% yield, 56% ee) and (*R*)-ibuprofen ethyl ester ((*R*)-(−)-**2b**, 32 mg, 32% yield, 67% ee); (*S*)-ketoprofen ((*S*)-(+)-**3**, 25 mg, 23% yield, 69% ee) and (*R*)-ketoprofen ethyl ester ((*R*)-(−)-**3b**, 55 mg, 45% yield, 25% ee); (*S*)-flurbiprofen ((*S*)-(+)-**4**, 29 mg, 28% yield, 97% ee) and (*R*)-flurbiprofen ethyl ester ((*R*)-(−)-**3b**, 61 mg, 52% yield, 51% ee). For additional data, see Table 5 in the main manuscript. For the results of specific rotation values for the EKR products, please see Table S2 appended in the Supporting Information.

## 4. Conclusions

The enantiomers of non-steroidal anti-inflammatory drugs have different pharmacodynamic and pharmacokinetic properties and hence have different therapeutic efficacies and safety profiles. Moreover, the metabolic configurational inversion of most of the administered NSAIDs, leading to biotransformation of the respective distomers into the corresponding eutomers, is minimal in humans. Both these phenomena provide a rational desire for using the single (*S*)-enantiomers of these drugs in therapy to reduce the total dose and toxicity that arises from non-stereospecific interactions with the (*R*)-enantiomers. Therefore, in this study, we focused our attention on the elaboration of a novel biocatalytic methodology that would extend the chiral synthetic toolbox for the preparation of enantiomerically enriched (*S*)-NSAIDs. In this regard, five different racemic 2-arylpropanoic acid derivatives were subjected to lipase-catalyzed kinetic resolution using trialkyl orthoesters as irreversible alkoxy group donors under mild reaction conditions (1 atm, 40 °C and 800 rpm). Preliminary analytical-scale studies allowed us to obtain desired eutomers with a moderate-to-high enantiomeric purity—(*S*)-flurbiprofen (97% ee), (*S*)-ibuprofen (91% ee), (*S*)-ketoprofen (69% ee) and (*S*)-naproxen (63% ee), respectively. In the preparative-scale EKRs, two of them containing secondary carboxylic groups gave promising results with *Candida antarctica* lipase B, thus furnishing (*S*)-enantiomeric forms of flurbiprofen in 28% yield and 97% ee, ibuprofen in 45% yield and 56% ee, whereas the other two have been synthesized in a moderate enantiomeric purity affording (*S*)-ketoprofen in 23% yield and 69% ee, and (*S*)-naproxen in 42% yield and 57% ee, respectively. In turn, racemic etodolac possessing a tertiary stereogenic center was unsuitable for the employed biocatalytic system, giving no reaction. Finally, orthoesters used in the enzymatic synthesis of (*S*)-NSAIDs seem to be a promising alternative to short-chain aliphatic alcohols.

**Supplementary Materials:** The following are available online at https://www.mdpi.com/article/10.3390/catal12050546/s1. Table S1. List of commercial enzyme preparations employed in these studies; Table S2. The results of specific rotation values for the EKR products; Table S3. HPLC analytical separation conditions of NSAIDs and their esters by chiral columns—(*S*,*S*)-Whelk-O 1 or Chiralcel OJ-H or Chiralpak AD-H. Next, the copies of HPLC chromatograms of NSAIDs and their respective esters in their racemic and optically active form, as well as copies of NMR, MS, FTMS, and IR spectra, can be found. References [45,84–86] are cited in the supplementary materials

**Author Contributions:** Conceptualization, P.B.; Methodology, B.Z., P.C., J.K. and P.B.; Software, P.B.; Validation, P.B.; Formal Analysis, P.B.; Investigation (experimental design and development of analytical methods), P.B.; Investigation (performing syntheses and analyses), B.Z., P.C. and P.B.; Data Curation, P.B.; Writing—Original Draft Preparation, P.B.; Writing—Review and Editing, B.Z., J.K. and P.B.; Visualization, P.B.; Supervision, P.B.; Project Administration, P.B.; Funding Acquisition, J.K. and P.B. All authors have read and agreed to the published version of the manuscript.

**Funding:** This research was financially supported by the National Science Centre (NCN) of Poland in the framework of grant SONATA 15 (No: 2019/35/D/ST4/01556).

**Data Availability Statement:** The data presented in this article are openly available.

**Acknowledgments:** B.Z. is grateful to the IDUB project ('Scholarship Plus' program) for providing a research fellowship. We acknowledge the Core Facility for Crystallography and Biophysics (supported by the Foundation for Polish Science under the European Regional Development Fund, TEAM TECH Core Facility POIR.04.04.00-00-31DF/17) for its valuable support. Statutory support by the Faculty of Chemistry at Warsaw University of Technology is also acknowledged.

**Conflicts of Interest:** The authors declare no conflict of interest.

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
