# Peer review of "Expanding Access to Optically Active Non-Steroidal Anti-Inflammatory Drugs via Lipase-Catalyzed KR of Racemic Acids Using Trialkyl Orthoesters as Irreversible Alkoxy Group Donors"

_catalysts, doi:10.3390/catal12050546_

Round 1

Reviewer 1 Report

    This article presented the selection of the commercial lipase for the resolution of 2-arylpropanoic acids, using various trialkyl orthoesters as donors. Most attractively, the enantiomeric purity of 97% ee toward (S)-flurbiprofen was achieved. This manuscript is well organized and clearly written. However, I still have some concerns which I list below

1. The results of enantiomeric purity stated in the abstract (lines 27 to 28), conclusion, Table 5, and part 3.6 should be checked carefully. There’s a differential data between the data in the abstract and that in Table 5 (or table 4). See, (S)-naproxen (63% ee)?

2. In the abstract or other parts of the text, the symbol of ee was carelessly used. However, this study showed the resolution of rac-1 to rac-4 with different parameters of enantiomeric purity by ees and eep. It should be clearly noted the symbol of ees or eep.

3. Line 164, “keeping the enzyme/substrate ratio close to 1:2 w/w” was not standard. According to 3.2, “naproxen (20 mg, 87 μmol) or ibuprofen (rac-2, 18 mg, 87 μmol)]”, “alkoxy group donor trialkyl orthoester [0.26 mmol; for trimethyl orthoacetate (Alk-D1): 31 mg…, Novozym 435-STREM (10 mg) were added”, both naproxen and alkoxy group donor trialkyl orthoester were the substrates for lipase Novozym 435-STREM. Therefore, the enzyme/substrate ratio should be redefined.

4. Table 1 showcased the several commercial lipases for the resolution of rac-1 and rac-2, the same amount of enzyme was applied. What was the activity for each commercial lipase? It was suggested to refer to the activity of lipase.

5. Lines 173 to 174, it claimed that “The obtained EKR products were isolated and purified using preparative column chromatography before being subjected to further HPLC analysis”. What’s the purpose of this procedure? Did it affect the enantiomeric purity results using isolation and purification by column chromatography for the HPLC measurement? How about the results determined by HPLC without isolation and purification by column chromatography? Moreover, what was the detailed method for isolation and purification by column chromatography? At least, a reference should be added.

6. A high conc. of rac 1-4 was used according to table 5, could 0.43 mM rac 1-4 absolutely dilute in the solvent system? Also, 10 mg Novozym 435-STREM was used for resolution reaction with high conc. of substrate? For the preparative-scale lipase-catalyzed reaction, why do not keep the same enzyme/substrate ratio as previous results?

Reviewer 2 Report

This study presents excellent results related to the application of the technique of enzymatic resolution of racemic mixtures. In this case it is obtained the separtion of boths enentioisomeros of the 2-arylpropanoic acids. This methodology is of great interest in organic synthesis, and can serve as inspiration for other processes of interest. The excellent bibliographic review included in the Introduction section is noteworthy. Therefore this manuscript can be accepted for publication in its present form.

Reviewer 3 Report

Paper entitled „Expanding Access to Optically Active Non-steroidal Anti-inflammatory Drugs via Lipase-catalyzed KR of Racemic Acids Using Trialkyl Orthoesters as Irreversible Alkoxy Group Donors” presents a valuable synthetic work applying lipase catalysts. Results about the wide substrate panel and the optimization of the lipase catalyzed kinetic resolution provide useful information for researchers from the field of organic chemistry and pharmaceutical development.

The manuscript is well structured and written. The introduction shows the background of this study quite well and the experimental part is more or less clear enough.

Only some minor issue and comment can be added to the manuscript:

  1. Authors should highlight the novelty of their work and a systematic design of experiments should be also useful to present at the end of Introduction.
  2. On Scheme 1, the definition of the applied organic solvent needs to be added on the reaction arrows.

  3. In Table 1, different form of immobilized CaLB can be found, which were applied for the KR of rac-1 and rac-2. The conversion rates are very difference depends of the type of immobilized enzyme’s, however all of them contain the same enzyme. Authors should add more details and explanation about these remarkable differences considering different aspects (for example enzyme loading, morphology, pore size, surface properties of the different carriers.)
